



# Historical trends of seasonal droughts in Australia

Matthew O. Grant[1,2], Anna M. Ukkola[1,2,3], Elisabeth Vogel[2,4,5], Sanaa Hobeichi[1,3], Andy J. Pitman[1,2], Alex Raymond Borowiak[2,5] and Keirnan Fowler[6]

[1]Climate Change Research Centre, UNSW Sydney, Kensington, NSW 2052, Australia
[2]Australian Research Council Centre of Excellence for Climate Extremes
[3]Australian Research Council Centre of Excellence for 21st Century Weather
[4]Water Research Centre, UNSW Sydney, Kensington, NSW 2052, Australia
[5]School of Geography, Earth and Atmospheric Sciences, University of Melbourne, Parkville, VIC 3052, Australia
[6]Department of Infrastructure Engineering, University of Melbourne, Parkville, VIC 3052, Australia

*Correspondence to*: Matthew O. Grant (matt.grant@unsw.edu.au)

**Abstract.** Australia frequently experiences severe and widespread droughts, causing impacts on food security, the economy, and human health. Despite this, recent research to comprehensively understand the past trends in Australian droughts is lacking. We analyse the past changes in seasonal-scale meteorological, agricultural and hydrological droughts – defined using the 15th percentile threshold of precipitation, soil moisture, and runoff, respectively. We complement these traditional metrics with an impact-based drought indicator built from government drought reports using machine learning. Calculating trends in time and area under drought, for the various drought types, we find that while there have been widespread decreases in Australian droughts since the early 20th century, extensive regions have experienced an increase in recent decades. However, these recent changes largely remain within the range of observed variability, suggesting they are not unprecedented in the context of the historical drought events. The drivers behind these drought trends are multi-faceted and we show that the trends can be driven by both mean and variability changes in the underlying hydrological variable. Additionally, using explainable machine learning techniques, we unpick the key hydrometeorological variables contributing to agricultural and hydrological drought trends. The influence of these variables varies considerably between regions and seasons, with precipitation often shown to be important but rarely the main driver behind observed drought trends. This suggests the need to consider multiple drivers when assessing drought trends.

## 1 Introduction

Droughts are one of the most damaging extreme weather events (Wilhite, 2000). Large impacts from droughts are felt across various sectors including economic damage, ecological degradation, and the loss of human lives (Bond et al., 2008; Cravens et al., 2021; Douris and Kim, 2021; Zaveri et al., 2023). They can threaten water supplies and food security (Peterson et al., 2021; Vogel et al., 2019), and have the potential to increase the risk and severity of heatwaves and wildfires (Adams et al., 2020; Jyoteeshkumar reddy et al., 2021). Australia is naturally prone to widespread and severe droughts (Falster et al., 2024). For example, during 2017-19, southeast Australia faced its most severe drought since reliable records began (Devanand et al., 2024). This drought led to severe threats to Sydney's water supply, intense agricultural impacts, and culminated in the



unprecedented and devastating 2019/20 Black Summer bushfires, which burnt 5.8 million hectares across southeast Australia (Abram et al., 2021).


Droughts have been changing regionally around the world (Seneviratne et al., 2021). However, this change is not uniform globally - for example, there is evidence for increased droughts in South America, but decreased droughts in northern Europe (Seneviratne et al., 2021). It is therefore important to better understand how droughts have been changing at regional scales. Despite this, the historic changes in droughts across Australia are not well understood. Past studies have considered the mean

changes to hydrological variables including rainfall (Dey et al., 2019; Taschetto and England, 2009) or metrics such as the Palmer Drought Severity Index (PDSI) or the Standardised Precipitation Index (SPI) (Dai, 2011; Rashid and Beecham, 2019; Yildirim and Rahman, 2022). However, metrics like the PDSI and SPI depict both wet and dry periods, and as such changes in these only quantify trends in the mean states, ignoring changes in variability. Droughts are anomalously dry periods driven by both the mean and variability and it is important to consider changes to both of these aspects when quantifying drought

trends (Ukkola et al., 2020). This is particularly true for Australia, which has extremely high natural rainfall and streamflow variability (King et al., 2020; McMahon et al., 1987; Nicholls et al., 1997).

Evidence suggests that mean precipitation has been increasing across most of the Australian continent since the early 20[th] century (Ukkola et al., 2019), but there has been a decreasing trend across the southeast and southwest since the middle of the

century (Dey et al., 2019). These decreasing trends from the mid-century are reflected in the streamflow and soil moisture over many areas (Dai, 2011; Wasko et al., 2021; Zhang et al., 2016), suggesting possible changes in droughts. Previous research that has considered the observed changes to anomalously dry periods, for both meteorological and agricultural droughts, across Australia showed that drought frequency, duration and severity had been decreasing since 1911 across most of the continent, with some exceptions in the southwest and southeast (Gallant et al., 2013). However, changes over the last decade, which

encompasses major drought events (e.g. Devanand et al., 2024), have not been considered as this data was not available at the time. The areal extent of hydrological droughts has been increasing in southwest and southeast Australia, and decreasing in the north and central regions of the country since 1960 (Wasko et al., 2021). While this study was able to capture recent hydrological drought trends, it only focussed on one aspect of these droughts (areal extent) and only considered trends from 1960. Most studies have only considered one or two drought types and use different metrics, making a comparison across

studies challenging. To gain a complete picture of historical drought trends, it is important to consider changes in multiple drought types, from meteorological to agricultural and hydrological drought (Cook et al., 2020; Kirono et al., 2020), using metrics which quantify anomalously dry periods (Ukkola et al., 2020).

Here we investigate changes in meteorological, agricultural and hydrological droughts across Australia during 1911-2020. We

compliment the traditional drought metrics with an impact-based drought metric, which has been built by training a machine learning model on government drought impact reports. By considering these four different drought metrics, we provide a more



complete overview of how Australian droughts have been changing over the historical period than previous studies have achieved. We additionally quantify the contribution of mean and variability changes to the drought trends and identify the key hydrometeorological variables contributing to agricultural and hydrological droughts.

## 2 Data and Methods

Here, we first introduce the hydrometeorological and climate datasets used in the analysis (Section 2.1). We then describe the methods used to define the drought metrics (Section 2.2) and calculate the drought trends (Section 2.3). To better contextualise these trends, we outline the approach for determining their emergence from long-term variability (Section 2.4). Finally, we explain the methodology used to quantify the contributions of changes in the mean and variability to the drought trends (Section 2.5), as well as the techniques used to assess the importance of various hydrometeorological variables to agricultural and hydrological drought trends (Sections 2.6).

### 2.1 Data

Table 1 provides all the data used in this study and identifies which part of the analysis the variable was used in (further discussed in following sections) and where the data was sourced from.

Precipitation was derived from the Australian Gridded Climate Data (AGCD) version 1 (Jones et al., 2009). AGCD is a gridded product of observed precipitation across Australia produced by interpolating observed station precipitation onto a $0.05° \times 0.05°$ grid. The network of observation stations is sparse in very remote regions, making the interpolation methods unreliable in these areas (Vogel et al., 2021). Therefore, for our analysis, we have masked out grid cells in these regions.

Root zone soil moisture (top 1m) and total runoff were obtained from the Australian landscape water balance model (AWRA-L; Frost et al., 2018) as continent-wide observations for these variables do not exist. AWRA-L is a semi-distributed, hydrological model covering the whole of Australia, which underpins the Bureau of Meteorology's Australian Water Outlook (https://awo.bom.gov.au). It is used to produce hydrological information on a range of timescales; from past hydrological conditions (Wasko et al., 2021), to seasonal forecasts (Pickett-Heaps and Vogel, 2022; Tian et al., 2021; Vogel et al., 2021) and future projections (Peter et al., 2024; Wilson et al., 2022). AWRA-L has been calibrated to observed streamflow, satellite soil moisture and evapotranspiration across Australia (Frost et al., 2018) and evaluated using various hydrological observations, including in-situ measurements of soil moisture, gauged streamflow, groundwater recharge, and flux tower-based evapotranspiration (Frost and Wright, 2018). AWRA-L data is on the same grid as AGCD so allows for direct comparison between the two datasets. For a full description of AWRA-L version 6 and its evaluation, see Frost et al. (2018) and Frost and Wright (2018). Streamflow observations from the Australian edition of the Catchment Attributes and





Meteorology for Large-sample Studies Version 2 (CAMELS-AUS v2; Fowler et al., 2024) were used to evaluate AWRA-L simulated hydrological droughts. This dataset provides streamflow observations at 561 river catchments across Australia.

There is no available dataset of gridded evapotranspiration (ET) observations, and as such we have used ET data from the Global Land Evaporation Amsterdam Model (GLEAM) version 3.6 (Martens et al., 2017). GLEAM calculates ET through a combination of remotely sensed observations and reanalysis data (for variables such as soil moisture, air temperature, and radiation). GLEAM data has been rigorously validated against in-situ evaporation observations and deemed to perform adequately (Martens et al., 2017). The GLEAM data is on a $0.25° \times 0.25°$ grid, and as such was regridded to the AGCD grid,
using the nearest neighbour method, to allow for direct comparison with the other gridded data products.

**Table 1: Data used in the study. RF drought trend analysis refers to the random forest (RF) analysis of contributions from hydrometeorological variables to drought trends (see Section 2.4 of methods and 3.3. of results for this).**

| Variable | Use | Time Period | Data Source |
|---|---|---|---|
| Precipitation | Meteorological drought metric<br>Impact-based drought metric<br>RF drought trend analysis | 1911-2020 | AGCD v1 (Jones et al., 2009) |
| Soil moisture (root zone) | Agricultural drought metric<br>Impact-based drought metric<br>RF drought trend analysis | 1911-2020 | AWRA-L v6 (Frost et al., 2018) |
| Total runoff | Hydrological drought metric<br>Impact-based drought metric<br>RF drought trend analysis | 1911-2020 | AWRA-L v6 (Frost et al., 2018) |
| Nino3.4 | Impact-based drought metric | 1911-2020 | NOAA (NOAA, 2021) |
| Southern Oscillation Index | Impact-based drought metric | 1911-2020 | Bureau of Meteorology (Bureau of Meteorology, 2024b) |
| Indian Ocean Dipole | Impact-based drought metric | 1911-2020 | NOAA (Kumar et al., 2020) |
| Evapotranspiration | RF drought trend analysis | 1981-2020 | GLEAM v3.6 (Martens et al., 2017) |
| Streamflow | Hydrological drought evaluation | 1951-2020 | CAMELS-AUS v2 (Fowler et al., 2024) |





## 2.2 Drought metrics

### 2.2.1 Traditional drought metrics

We have used metrics for three common drought types, referred to as traditional drought metrics, describing meteorological (precipitation), agricultural (soil moisture) and hydrological (runoff) droughts. To identify drought months, the hydrological variables were averaged using 3-month running means so that the new value at any given month was the average of that month and the two preceding months. A drought threshold for each month was then set as the 15th percentile of the study period
(1911-2020). A drought month was identified when the 3-month mean was below its corresponding 15th percentile threshold. We repeated this for annual-scale drought by using 12-month running means. We focus on the 3-month (seasonal) droughts in the main paper but present results for 12-month (annual) droughts in the supplementary information.

Seasonal droughts were chosen as they impact multiple sectors in Australia including agriculture, water resources and
environmental systems (Gallant et al., 2013; Ukkola et al., 2024). Additionally, by looking at droughts on this time scale, we can determine the changes in individual seasons. We chose to use the 15th percentile as the threshold limit as this is approximately equal to an SPI threshold of −1 (i.e. a "moderate" drought; Mckee et al., 1993). Additionally, considering seasonal droughts at the 15th percentile ensures a large enough sample size of drought events to reliably calculate trends. By using an empirical percentile method, no assumption of a specific statistical distribution of the hydrological variable is required,
making it suitable for application across multiple drought indicators. This chosen method is consistent with previous studies (Ukkola et al., 2020, 2024) and similar to the definition used by the Australian Bureau of Meteorology (Bureau of Meteorology, 2024a).

### 2.2.2 Impact-based drought metric

To develop the impact-based drought metric, a Random Forest (RF) binary classification algorithm (Breiman, 2001) was used
to model the relationship between observed drought impacts and climate conditions, as in Devanand et al. (2024). Various iterations of the RF model were created, using multiple aggregations of climate predictor variables, with the final version being optimised for highest performance when classifying unseen drought months. The final RF model uses six climate variables, ranging from large-scale modes of variability to localised climate conditions, and the month of the year as predictors. Table 1 provides details of the climate variables used in the final model. The observed drought impacts data is a database of months
experiencing drought impacts reported by Australia's Bureau of Meteorology, New South Wales (NSW) Department of Primary Industries, and NSW Department of Planning, Industry & Environment. These were balanced by an equal number of months of "no-drought" events to allow for the RF model training (see Fig. S1 for further details of location and time of these reports).  RF models trained on drought impact reports have been shown to perform well for classifying drought events (Devanand et al., 2024; Hobeichi et al., 2022), outperforming traditional drought metrics for drought prediction and capturing





nonlinear or compounding relationships between climate variables and drought events which linear models might struggle to represent (Hobeichi et al., 2022). As such, we have chosen to use RF models to construct our impact-based drought metric.

The performance of the RF model was assessed through out-of-sample testing. For this, 70% of the reported drought events were used to train the RF model, with the other 30% withheld to be used as test data. This was repeated 100 times, each time

creating a new RF model on a new random 70/30 split of the data. The performance of the RF model was assessed by aggregating the performance of each RF model on its 30% out-of-sample data. Five performance metrics were used to assess the RF models: accuracy, precision, recall, balanced accuracy, f-1score, and false alarm rate. These are all commonly used in binary classification performance assessment and similar to those used by Hobeichi et al. (2022). The results of these performance metrics can be found in Fig. S2.


Once the model had been tested and evaluated, the impact-based drought metric was developed across southeast Australia from 1911-2020 using all the available data (i.e. without withholding test data). The impact-based drought metric was developed for southeast Australia as the drought-impact reports used in the training of the RF model are only available for this region. Note that this metric identifies drought months but does not provide information on the intensity of drought events.

**2.2.3 Drought characteristics**

Three characteristics of droughts are considered in this study: time under drought, area under drought, and drought intensity. Time under drought is calculated for all four drought types including the impact-based metric, whereas area under drought and drought intensity are calculated for the three traditional drought metrics only.

The time under drought was calculated from the binary timeseries of drought months (see Section 2.2.1 and 2.2.2) by summing the number of drought months per grid cell over distinct 5-year time blocks. Temporal resampling in 5-year blocks was done to create a continuous timeseries from which trends can be calculated while ensuring that each block is long enough to include drought and non-drought events. Multiple aggregation periods (2-, 3- and 7-year blocks) were also tested, and they had little effect on the results.


The area under drought was defined as the percentage of grid cells under drought at each timestep (using the binary drought timeseries for each grid cell, see Section 2.1.1). Given the variation in climate conditions across Australia, this metric was calculated over the eight Natural Resource Management (NRM) regions (Fig. 1). These NRM regions represent broad regions of similar climate conditions and biophysical factors (CSIRO and Bureau of Meteorology, 2015).



Drought intensity was defined as the relative deviation from the long-term mean. First, drought events were identified as consecutive months for which the relevant hydrological variable was below the 15[th] percentile threshold. The intensity was then calculated as the percentage difference between the climatological mean and the mean of the variable across all months for which the event lasted. By calculating the percentage difference rather than absolute differences, we were able to compare between drought types and across locations.

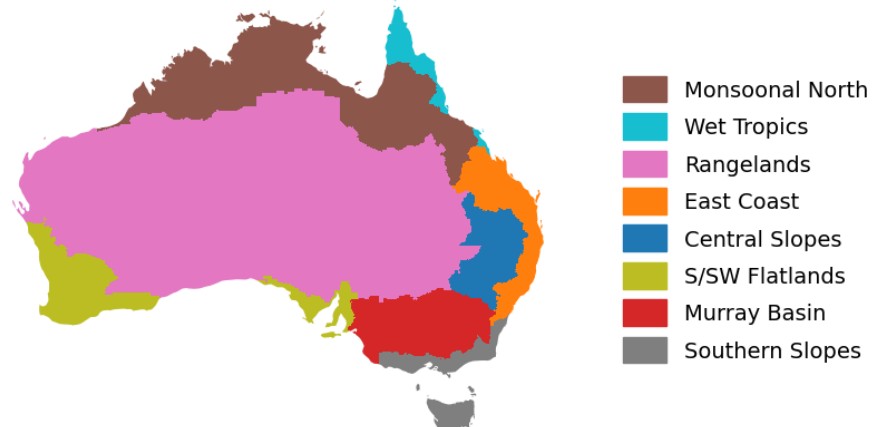

**Figure 1: Natural Resource Management (NRM) regions.**

## 2.3 Drought trends

The Mann-Kendall (MK) trend test and Theil-Sen (TS) slope estimator (Sen, 1968) are common nonparametric methods for respectively calculating the significance and slope of trends in hydrometeorological time series (Deitch et al., 2017; Humphrey et al., 2016; Zhang et al., 2016). The MK trend test has been shown to not perform well when applied to autocorrelated data

(Yue et al., 2002), so here we apply a modification of the MK trend test which addresses this issue, proposed by Yue and Wang (2004). This modification of the MK trend test was used to calculate the significance and direction of the trend in each of the drought characteristics. The trend slope was then calculated using the TS slope estimator. In rare cases, the MK trend test would detect a significant trend in the data, but the TS slope estimator would find the gradient of the trend slope to be zero. This occurs when the data contains many zero values. In these cases, the significance of the trend was set to be 'insignificant'

to keep consistency between the significance and slope of the trend.

To ensure that hydrological drought trends are reliable, we evaluated them against observed hydrological drought trends based on in-situ streamflow data from CAMELS-AUS v2. Details of this verification can be found in Section S1.1 of the Supplement.

## 2.4 Trend emergence tests

To better understand whether the trends are within the expected range of the historical variability, trend emergence tests were conducted. Firstly, for area under drought trends, we analysed whether the changes between decadal means are statistically





significant when compared to the variability over the historic period. Adopting the methods used in previous studies (Sun et al., 2018; Ukkola et al., 2019), we calculated 95% confidence intervals for the decadal means. To do this, first the lag-1 autocorrelation, $r_1$, was calculated by:


$$r_1 = \frac{\sum_{i=1}^{n-1}(Y_i - \bar{Y})(Y_{i+1} - \bar{Y})}{\sum_{i=1}^{n}(Y_i - \bar{Y})^2} \qquad (1)$$

where $Y$ is the timeseries; $Y_i$ is the data at timestep $i$; $\bar{Y}$ is the mean of the timeseries; and $n$ is the total number of timesteps. This was then used to calculate the effective sample size, $n_e$:

$$n_e = n\left(\frac{1 - r_1}{1 + r_1}\right) \qquad (2)$$

And finally, from this we could find the confidence intervals with:

$$CI_{decadal} = \pm 1.96 \sqrt{\frac{\sigma^2}{n_e} + \frac{\sigma^2}{10}} \qquad (3)$$

where $\sigma^2$ is the variance of the whole time series. If the decadal means of area under drought are within these confidence intervals, it suggests that the changes in the area under drought remain within the expected range of the historic variability. If the decadal means emerge outside the confidence intervals and remain outside until the end of the timeseries, then it is possible that the trend has emerged from the variability. Additionally, the signal-to-noise (S/N) ratio and Kolmogorov-Smirnoff (KS) test were used to determine if time and area under drought has emerged from their historical variability (see Section S1.2 in

the Supplement for details). Both these tests are widely used methods to test for emergence (e.g. Hawkins et al., 2020; King et al., 2015).

**2.5 Contributions from mean and variability changes**

The drought trends identified in this study could be influenced by mean and variability changes to the relevant hydrological variable. The contribution of these mean and variability changes was assessed for trends in time under drought for each of the

traditional drought types. To calculate the contribution from changes in variability, the long-term trend was removed from the relevant hydrological variable using linear detrending. Drought months were then recalculated on this detrended hydrological variable, and the trend in time under drought of this new drought metric was calculated. As we have removed the change in the mean of the hydrological variable, this new trend is caused solely from the change in variability of the hydrological variable. We refer to this as the variability drought trend. The relative contributions from the variability and mean changes of the

hydrological variable were then computed as:



$$variability\ contribution = \frac{variability\ drought\ trend}{original\ drought\ trend} \times 100 \qquad (4)$$

$$mean\ contribution = 100 - variability\ contribution \qquad (5)$$

Often the contributions would act in opposite directions. For example, the variability drought trend may be negative when the original drought trend was positive. This would give a negative variability contribution, and a mean contribution greater than 100%. In these cases, we defined the drought trend to be purely caused by changes in the mean, and as such the variability contribution was set to 0% and the mean contribution was set to 100%. The same logic was applied when the variability

contribution was initially found to be above 100% and the mean contribution negative: in these cases, the variability contribution was set to 100% and the mean contribution to 0%. Therefore, the final values for contributions of mean and variability change are in the range of 0-100%.

**2.6 Contribution of hydrometeorological variables**

We identified the contribution of various hydrometeorological variables to agricultural and hydrological drought trends to

identify the extent to which they are driven by precipitation versus other influences. These were assessed by linking drought trends to trends in hydrometeorological predictors using an RF model as a regression algorithm (Breiman, 2001). The influence of precipitation, ET, and runoff on agricultural drought trends, as well as the influence of precipitation, ET, and soil moisture on hydrological drought trends, were evaluated. Further details of these variables can be found in Table 1. For each variable, its trend, as well as the trend in its standard deviation, from 1981-2020 was calculated using the TS slope estimator. 1981-

2020 was chosen as this was the longest time period for which all predictor data was available. A separate RF model was trained for each season and NRM region, with the 1981-2020 time under drought trend as the target variable and the trends in the hydrometeorological variables, along with the trends in their standard deviation, used as the predictors.

Before implementing the RF models in this analysis, their ability to capture the drought trends was tested. Each RF model was

assessed by withholding 30% of the drought trend data to be used for out-of-sample testing. We iterated this 100 times with a new model trained on a new random 70/30 split of the data each time. The performance of the RF model was then assessed by finding the $R^2$ score between the observed out-of-sample drought trends and the respective predicted drought trends. The $R^2$ score for each RF model can be found in Table S1. These scores varied depending on region and season but were in the range of 0.46 to 0.84 (mean of 0.69) for agricultural drought trends and 0.38 to 0.70 (mean of 0.56) for hydrological drought trends.

Due to this, it was deemed that the models performed adequately to be used in this analysis.

Once the models had been tested, the final analysis was conducted by training the models on all the available data. For each season and NRM region, 100 models were trained with a different random seed for each iteration. The variable importance





feature of RF models was used to assess the relative contributions of the predictor variables, with the results shown as the
mean importance ranking of the 100 models. Here, we use the Mean Decrease in Impurity (MDI) variable importance method
(Breiman, 2001), with higher MDI scores indicating greater importance. RF models also allow for the assessment of variable
importance through the permutation importance method. However, MDI is better at handling predictors which are highly
correlated and as such is more appropriate for our analysis. For each model, a random variable was generated and added as a
predictor to give a baseline comparison for the importance scores of the other predictor variables. While the importance scores
do not measure causal relationships, the predictors which rank highly will likely have a substantial influence on drought trends.

## 3 Results

### 3.1 Time under drought trends

To understand if Australia has experienced a change in drought events, we examined the observed trends in time under drought.
The trends were calculated for each of the three traditional metrics over the periods 1911-2020, 1951-2020, and 1971-2020 to
assess how the trends have evolved over the historical period (Fig. 2). During 1911-2020 the time under drought is decreasing
across the large majority of Australia. There are particularly large areas in the northwest of the country that are showing
consistent and significant decreasing trends. Increasing drought trends are apparent in the southwest for the three drought
types, as well as some regions in the east coast, southeast and Tasmania for agricultural and hydrological drought.





Figure 2: Trends in time under drought for the three traditional drought types and three time periods. The maps show the change in the number of drought months per 5 years during the three time periods. The hatching indicates where the trend is not significant (p > 0.05). The white spaces indicate the area masked out due to sparse observation network.

Over 1951-2020, there are much larger areas of increasing trends in time under drought. This is particularly evident over the eastern half of the country, and along much of the west coast. However, large areas in the north and northwest of the country still show significant reductions in time under drought. The areas of increasing trends are even larger for 1971-2020, with most of the country experiencing drying trends over this period. Many areas in central and northern Australia are showing increasing time under drought compared to decreases over the longer time period. Although some areas in the north are still showing decreasing trends, these areas are much smaller than for the 1951-2020 period. These trends are much stronger than what was seen over 1911-2020, with meteorological and agricultural droughts sometimes showing an increase of three or more extra



drought months per five years, and hydrological droughts sometimes increasing by over six extra drought months per five years.

Although these trends are often significant, there are few areas where the trend has emerged from the historic variability as indicated by the KS test and S/N ratio (Fig. S3). The S/N ratio shows no areas where the trend has emerged for meteorological

and agricultural drought, and only for 0.02% of the country for hydrological drought. For meteorological and agricultural drought, the KS test also only shows small areas (0.8% and 0.6%, respectively) where the trend has been found to have emerged. However, for hydrological drought, around 15.3% of the country is showing emerging trends. When these changes are showing an increase, much of this is concentrated in the southwest, consistent with a strong decline in streamflow in this region (Petrone et al., 2010). There is also a substantial region of increasing emerged change near the east coast. However,

most of the area (14.5% of the country) showing an emerging trend is experiencing a decreasing trend. This is largely concentrated in the north and northwest; areas which have been experiencing a substantial increase in streamflow in recent decades (Wasko et al., 2021). However, given the inconsistency between the two tests, there is not strong evidence to suggest that these trends are outside the variability of the observational period. This is in line with evidence from the paleoclimate record which shows the frequency of droughts in the southeast and southwest of Australia are within the natural variability of

the climate when compared to paleoclimate data and climate models (Falster et al., 2024; O'Donnell et al., 2021; Vance et al., 2015).

These trends are largely reflected in the impact-based metric across southeast Australia. The 1911-2020 period shows areas of decreasing time under drought across parts of inland southeast Australia, whereas increasing trends are seen along the east

coast (Fig. 3). Though, much of the region shows no significant changes. However, during 1951-2020 trends show increasing time under drought over many regions, particularly along the coast and in western Victoria. The increases become widespread over the 1971-2020 period, covering most of the region. However, these trends are rarely significant, likely due to high variability of the impact-based drought metric and the shorter time period over which these trends were calculated. The trends in the impact-based metric support the results found using the traditional drought metrics yielding similar results.







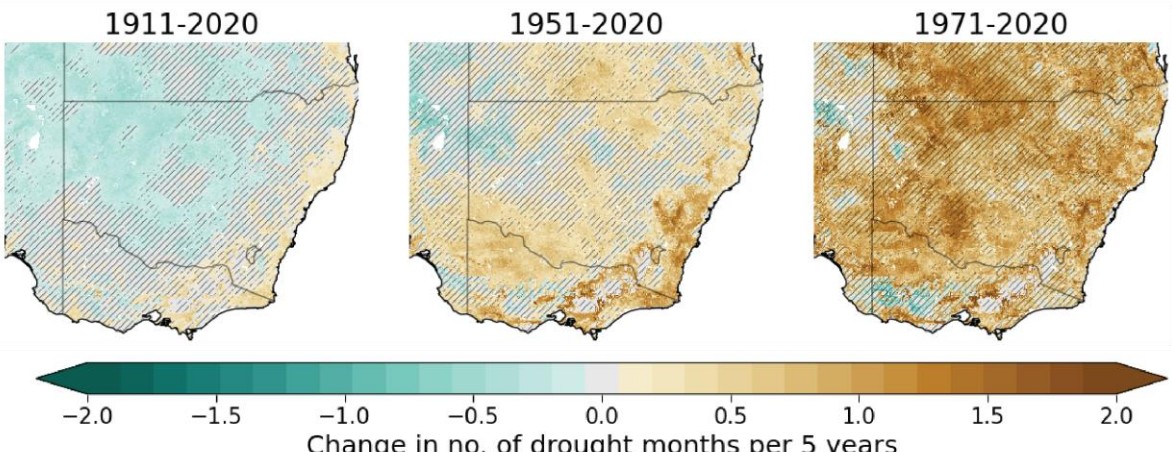

**Figure 3: Trends in time under drought for the impact-based drought metric. The maps show the change in the number of drought months per 5 years during the three time periods. The hatching indicates where the trend is not significant (p > 0.05). The white spaces indicate the area masked out due to sparse observation network**.

We tested the robustness of the trends detected in the traditional drought metrics using alternative methods. We aggregated the drought months over different periods, i.e. over 2, 3, and 7 years (Fig. S4-6); used a logistic regression model to model the changes in the drought months (Fig. S7); and calculated the metrics using 12-month aggregation periods instead of 3 (Fig. S8). All these methods give results which largely agree with those presented in the main paper. This suggests that our results are robust to our methodology. The trends in drought intensity indicate very similar patterns to the time under drought trends (Fig.

S9).

### 3.2 Area under drought trends

Next, we quantify the area under drought trends across the eight NRM regions (Fig. 1) for the three traditional drought metrics. This allows for a more in depth look at how droughts have been changing over the historical period. The timeseries of area under drought is plotted alongside the trends over each half of the time period (1911-1965, and 1966-2020) and the decadal

means (Fig. 4). There is often a substantial difference in the trends over the two time periods. Over the whole period, there is a clear decline in area under drought for the northern and central regions (Monsoonal North, Wet Tropics, and Rangelands). This decline is especially prominent in the second half of the timeseries in Monsoonal North and Rangelands, however the decline is more prominent in the first half of the timeseries for the Wet Tropics. The S/SW Flatlands is the only region showing a consistent drying trend over the two periods. These findings are consistent with evidence of a decrease in southwest

Australia's rainfall and an increase in droughts (Seneviratne et al., 2021), and an increase in rainfall in northern Australia (Dey et al., 2019). By contrast, the regions over the southeastern half of the country (East Coast, Central Slopes, Murray Basin, and Southern Slopes), show a change in the direction of the trend between the two periods. In the first half of the timeseries, they show a clear and often significant decreasing trend, but this changes to an increasing trend between 1966-2020. This is





consistent with our findings of time under drought trends, where we see large areas across eastern and southeastern Australia
begin to increase from around the mid-20th century.

These observed changes in area under drought, although often significant, largely lie within the range of historical variability.
The decadal averages in area under drought nearly always remain within the confidence intervals of expected decadal
variability (Fig. 4). An exception to this is in the S/SW Flatlands and Southern Slopes, where the decadal averages rise above
the confidence interval for hydrological droughts towards the end of the timeseries. However, for the Southern Slopes, the
decadal averages fall back within the confidence intervals after having emerged beyond them and, for the S/SW Flatlands,
only the final decadal average of the timeseries is outside the confidence intervals. As such, there is limited evidence that these
changes have emerged beyond the expected historical variability of area under drought. Additionally, the KS test and S/N ratio
show that the changes remain within the historic variability for both meteorological and agricultural drought (Fig. S10-11). On
the other hand, for hydrological drought, both tests show that the changes are outside the expected range for the Monsoonal
North and the Rangelands. However, these never extend beyond an S/N ratio of less than -2, which suggests that these levels
of area under drought are unusual, but not necessarily unfamiliar (Frame et al., 2019). As the decadal means for these areas
remain within the confidence intervals, there is again inconsistency between tests. As such, there is not strong evidence to
indicate that these changes fall beyond the range observed variability during the historic period.





**Figure 4: 5 year rolling mean of area under drought for the three traditional drought metrics over the NRM areas. The trend from 1911-1965 is shown in magenta, and the trend from 1966-2020 in red. The significance of these trends are indicated in the top left hand corner of each plot (p = N/S: not significant, p < 0.05: signifcant, p < 0.001: very significant). The black horizontal bars represent decadal means, and the light blue shading indicates the range of the decadal confidence intervals (Eq. 3).**





### 3.3 Drought trends per season

We next investigate drought trends over different seasons to understand implications for different sectors that depend on seasonal conditions such as agriculture. The trends in time under drought differ strongly between summer (DJF) and winter (JJA) (Fig. 5). There are also clear differences for spring (SON) and autumn (MAM) (Fig. S12). These differences are apparent
for all traditional drought types considered but the differences between seasons are much less distinct in hydrological droughts. For the 1911–2020 period, DJF shows substantially larger areas of decreasing time under drought than JJA. In fact, only hydrological drought shows substantial areas of decreasing trends in JJA (36% of the total area, compared to 15% and 19% for meteorological and agricultural droughts, respectively). The areas of decreasing hydrological drought trends are mostly concentrated in the northwest, whereas there are large areas of increasing meteorological and agricultural drought in the
southwest. On the other hand, in DJF, 38-51% of the area is experiencing decreasing drought, and only 2-5% is experiencing increasing trends. The areas of decreasing trends are largely concentrated in western, northern and southeastern regions.

As with the annual trends, more recent decades tend to show larger areas of increasing drought. However, in DJF, there are still large areas of decreasing drought (37-48%) during 1951-2020, mostly over the western parts of the country. Though, there
are also areas of increasing trends on the eastern side, particularly for agricultural and hydrological droughts. In the 1971-2020 time period, the areas of decreasing drought reduce substantially (11-18%). The areas of increasing drought are larger, though still mostly apparent on the east of the country. For JJA, during 1951-2020 there is little area of decreasing drought, particularly for meteorological and agricultural droughts (5% and 9%, respectively), and there are substantial areas (25-33%) of increasing trends, particularly in the southwest which receives the majority of its rainfall during the cool season (Potter et al., 2005). Over
1971-2020, the drying in the southwest is still apparent, but there is a large area of wetting over the eastern part of the south coast in meteorological and agricultural droughts. This area is seen to be drying during the other seasons and the annual trends.



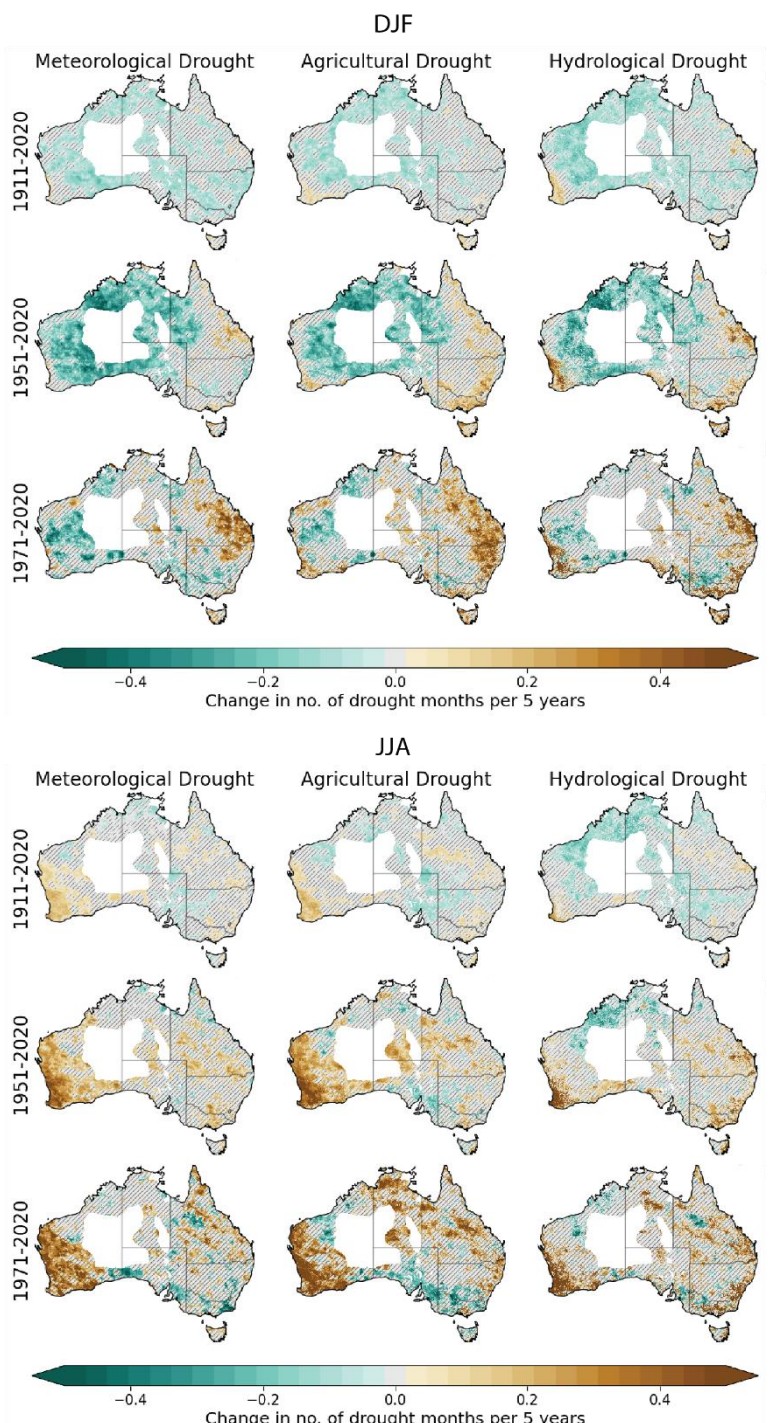

**Figure 5: Seasonal trends in time under drought for summer (DJF) and winter (JJA). The trend is shown for the three traditional**
**drought types and three time periods as the change in number of drought months per 5-years. The hatching indicates where the trend is not significant (p > 0.05). The white spaces indicate the area masked out due to sparse observation network.**





### 3.4 Variability and mean contributions

The trends in the drought metrics can be driven by changes in both the mean and variability of the underlying hydrological
variable. We next quantify these respective contributions for each drought metric to better understand the causes of past trends.
The contribution from the change in the mean is shown for the three traditional drought metrics (Fig. 6). A contribution of
100% signifies the drought trend is fully driven by changes in the mean and a contribution of 0% means the trend is fully
driven by changes in the variability. While there is some apparent randomness where the contribution is strongly contrasting,
there are large and coherent areas across the continent where the contribution from mean, and the contribution from variability,
is spatially consistent. In 78-92% of the area with a significant trend, depending on drought type and time period, the change
in the mean is the dominant cause of the drought trend. However, there are large areas (7-19% of the of significant trends)
where the change in variability is driving the drought trend. In many of these areas, the mean contribution is 0% and the
drought trend is being caused solely by the variability changes. In these cases, the trend of the underlying hydrological variable
is in the opposite direction to the drought trend. As such, if a study were to use mean changes as a proxy for drought changes,
which is often done (e.g. Dai, 2011; Feng and Zhang, 2015; Yildirim and Rahman, 2022), then the results would show trends
in the wrong direction for a substantial portion of the country. This highlights the importance of considering anomalously dry
periods when quantifying drought trends, as changes in the mean of hydrological variables may not always capture the trend
in the dry extremes correctly.





**Figure 6: The contribution of changes in the mean and variability of the underlying hydrological variable to trends in time under drought. This is shown for each of the three traditional drought types and for the trends over three different time periods. The white spaces indicate the area masked out due to sparse observation network. Grey indicates areas where the time under drought trend was not significant (p > 0.05).**

## 3.5 Contribution of hydrometeorological variables to drought trends

We next quantify the contribution of individual hydrometeorological drivers to drought trends. By relating trends in a number of hydrometeorological variables with the time under drought trends (using a RF as a regression model), we can identify those variables that have the strongest association with the drought trends. Note the RF method identifies correlation and not necessarily causation. However, we identify variables that have the strongest relationship with drought trends and are plausible drivers of these trends, thereby determining the most probable contributors to drought trends. Here, we focus on agricultural





and hydrological drought trends as these can be influenced by multiple aspects of the water cycle. The importance rankings of

the hydrometeorological variables for the 1981-2020 trends in time under agricultural (Fig. 7) and hydrological (Fig. 8) drought

for each NRM region over each season are presented.

**Figure 7: The variable importance of changes in key hydrometeorological variables for the 1981-2020 trends in time under**
**agricultural drought. These are shown for each NRM region and each season. The drought trend over the region and season is shown**
**in brackets in the legend of each plot with an asterisk indicating a statistically significant trend (p < 0.05).**

For the agricultural drought trends, we find large variations in the most important predictor variables depending on the region

and season. However, for all seasons and regions, every variable ranks higher than the randomly generated one, as such all

have some influence over the drought trends. The mean trend in ET is the variable most often ranked as the most important

across seasons and NRM regions. However, this varies greatly between the different regions; for example, the mean trend in

precipitation is the most important in the S/SW Flatlands for all seasons other than JJA. On the other hand, the Southern Slopes

 

is much more seasonally dependent: the most important predictor is the runoff mean trend in DJF and JJA, ET mean trend in MAM, and precipitation standard deviation trend in SON.

For the hydrological drought trends, again all variables are found to have influence over the trend as they rank above the random variable. The soil moisture mean trend is often the highest ranked variable across seasons and regions. One exception to this is the S/SW Flatlands, where the precipitation mean trend dominates in JJA and SON, while all the variables are ranked relatively equally in the other seasons.

**Figure 8: The variable importance of changes in key hydrometeorological variables for the 1981-2020 trends in time under hydrological drought. These are shown for each NRM region and in each season. The drought trend over the region and season is shown in brackets in the legend of each plot with an asterisk indicating a statistically significant trend (p < 0.05).**

Although precipitation trends are clearly an important factor in historical drought trends, precipitation only ranks as the most important variable in 16% and 22% of regions and seasons for agricultural and hydrological drought trends, respectively. This



emphasises the importance of not only using mean changes in precipitation as a proxy for drought. Droughts are often far more nuanced and can be heavily influenced by other land surface and hydrological processes. Agricultural and hydrological droughts are changing, and this is not solely attributable to precipitation changes, so it is imperative that future work considers the multiple factors which influence drought.

## 4 Discussion and Conclusions

### 4.1 Implications for drought impacts on agriculture and water supply

Across the southeast and southwest of Australia, we found that recent decades have been experiencing widespread increases in time and area under drought. While these changes likely remain within historic variability, they can still cause substantial impacts. Both the southeast and southwest are of particular importance to Australian agriculture with Australia's wheatbelt concentrated across the two areas (Vogel et al., 2021). The southeast produces around 40% of the country's agricultural output (Devanand et al., 2024). The time under drought is showing increasing trends in JJA and SON, which could have implications for winter cropping activities, such as the winter wheat industry. Historically, there have been severe drought impacts on agricultural production, particularly in recent decades. For example, wheat and barley production dropped by 73% and 43%, respectively, in 2018, during southeast Australia's 2017-19 Tinderbox Drought (Devanand et al., 2024). Similarly, dryland wheat production declined by an estimated 18-22% over 2002-2009, during one of southeast Australia's longest recorded droughts, known as the Millenium Drought (van Dijk et al., 2013). Future projections suggest that the recent historical trends could continue across southeast Australia (Kirono et al., 2020; Ukkola et al., 2024).

In addition to the agricultural impacts, droughts have major impacts on water supply. Over the more recent time periods (1951-2020 and 1971-2020), there have been increasing trends in time and area under hydrological drought for large areas in the southeast and southwest. The majority of the Australian population live within these areas, and so increased hydrological drought here could translate to impacts on water supply. Water scarcity has arisen during previous large droughts in these areas. In 2009, the final year of the Millenium Drought, Melbourne's water storage dropped to a quarter of its capacity, the lowest levels on record (Low et al., 2015), and water management practices were implemented to half the city's water consumption (Grant et al., 2013). Additionally, during the Tinderbox Drought, Sydney's water supply was severely threatened and many rural southeast Australian towns came close to running out of water (Devanand et al., 2024). With time under drought in southeast Australia projected to continue to increase (Kirono et al., 2020; Ukkola et al., 2024), Australia's major cities and rural townships could face further increased water scarcity risks.

### 4.2 Limitations

There are limitations stemming from the data used to identify the drought trends. The rainfall data, from AGCD is derived from gauge-based observations across the continent (Jones et al., 2009). These gauges are then interpolated across the country



to give a smooth surface at 0.05° resolution across the country. This method works well when the gauges are densely concentrated. However, it can cause issues when the gauges are spread far apart, as is the case in many parts of central Australia
(Chua et al., 2022). Even though the AGCD data has been evaluated comprehensively (Jones et al., 2009), the drought trends in data-sparse regions should be interpreted cautiously. These include the sparsely populated inland regions and the mountainous regions of eastern Australia in particular.

Similarly, AWRA-L brings its own uncertainties and errors despite the comprehensive evaluations which have been undertaken
to validate the hydrological model (Frost and Wright, 2020). One source of this uncertainty comes from its simplistic representation of the effect vegetation processes have on the water cycle and consequently evapotranspiration (Ukkola et al., 2024). Given these vegetation processes have been shown to have a large influence over water scarcity in Australia (Trancoso et al., 2017), this simplistic representation could lead to uncertainty in the model data. Additionally, AWRA-L does not account for human use of reservoir or aquifer water, which will have an influence of the water cycle. These areas of uncertainty in the
data will ultimately lead to uncertainties in our results. Our evaluation of AWRA-L showed good agreement against streamflow observations for the direction of hydrological drought trends (Figure S13). For the 1981-2020 period, AWRA-L runoff captures the correct sign of the trend (negative, positive, or zero) at 76% of the catchments, and 86% of catchments for the 1951-2020 trends despite lower agreement in the magnitude of trends. We note that the streamflow catchments are mainly concentrated along eastern coast of Australia with fewer gauges in the west and north of the country. From the information we
have in the other areas of Australia, the catchments on the west half of the country show a trend direction agreement of 76% between the model and observations, and 59% in the northern areas. This suggests that AWRA-L consistently captures the direction of the drought trend across the country, but performs slightly worse in the northern areas, where the wetting trends dominate. However, the robustness of this result is harder to evaluate for the western and northern areas due to lower observational data.

**4.3 Key Messages**

A comprehensive understanding of how Australian droughts have been changing has been missing from recent literature. Here, we have shown that across large areas of the country the occurrence of seasonal droughts has significantly declined over the past century with the clear exception of southwestern Australia. However, in more recent decades the time under drought has significantly increased across many regions, particularly across the east and southwest. We find similar changes in annual-
scale droughts. Notably, the decreasing trends over the past century are more apparent in summer (DJF), and the increasing trends in the latter half of the century are more pronounced in winter (JJA) and spring (SON). These changes have potential implications for changes to drought impacts. The steady increase over the second half of the 20[th] century could bring with it increased drought risk to various industries – such as agriculture and water supply - and the natural environment. However, this increase is following a decline in time under drought and has currently only rebounded to similar levels experienced in the

early 20th century.  As such, these recent changes are not unprecedented in the context of the last century, and it is unlikely that these changes extend beyond the historical variability of past droughts.

These drought trends are influenced by both the mean and variability changes in the underlying hydrological variables of each drought type. Even though the mean changes are the primary driver of significant drought trends across most of Australia,

there are large areas where the variability changes dominate (7-19% depending on the drought metric and time period), highlighting the need to consider variability changes when quantifying drought trends. We also show that, in many regions, the trends of agricultural and hydrological droughts are not dominated by changes in precipitation but strongly influenced by land surface processes including changes in evapotranspiration. These results highlight that simply evaluating changes in mean precipitation is not sufficient for quantifying trends in droughts.

**Competing Interests**


The authors declare that they have no conflict of interest.

**Author contribution**

MOG, AMU, and EV conceptualised the study. The specific methodologies were developed by MOG, AMU, EV, SH, and ARB. MOG and AMU carried out the analysis and developed the analysis code. All authors contributed to the interpretation

of the results. MOG wrote the first draft of the manuscript; all authors contributed to the final manuscript.

**Acknowledgments**

This study was funded by the Australian Research Council (ARC) Centre of Excellence for Climate Extremes (CE170100023). AMU is supported by the ARC Discovery Early Career Research Award (DE200100086). SH acknowledges the support of the ARC Centre of Excellence for the 21st Century Weather (CE230100012). This research was undertaken with the assistance

of resources from the National Computational Infrastructure (NCI Australia), an NCRIS enabled capability supported by the Australian Government.

**Code Availability**

The analysis codes are available at https://github.com/MattGrant1998/AUS_histroical_drought_trends.

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
