# Peer review of "Historical trends of seasonal droughts in Australia"

_EGUsphere, 2024_

## Author Comment (AC1)

**Reviewer 1**

**Review of Grant et al., Historical Trends of Seasonal Droughts in Australia**

This manuscript analyzes historical drought trends in Australia using multiple drought indicators, including meteorological, agricultural, hydrological, and impact-based metrics. It employs explainable machine learning to assess hydrometeorological drivers and the influence of variability versus mean changes. The study is well structured, the methodology is generally robust, and offers a valuable contribution to the understanding of historical drought trends in Australia. However, I have a several recommendations for improvement to enhance clarity, statistical transparency, and methodological rigor. This review evaluates the manuscript against HESS criteria, with recommendations to enhance its impact and alignment.

We would like to sincerely thank the reviewer for taking the time to review the manuscript and for their positive assessment.

**Scientific Significance**
**1. Does the paper address relevant scientific questions within the scope of HESS?**
- Yes, the manuscript examines historical drought trends and their drivers, contributing to the understanding of water availability and hydrological variability. The study's integration of multiple drought indicators and machine learning techniques is relevant to advancing hydrological monitoring and data analysis. Additionally, its focus on drought impacts aligns with HESS's interest in hydrology's interaction with climate and society.
- **Suggested Revision:** The discussion could further highlight how the findings might support sustainable water resource management and decision-making in response to drought variability.

Thanks for this suggestion. We will expand the discussion section to highlight the implications on water resource management and decision-making.

**2. Does the paper present novel concepts, ideas, tools, or data?**
- Yes, ML integration with traditional drought assessment is emerging in Australia (line 135, citing Devanand et al., 2024; Hobeichi et al., 2022). While advancements exist, the field is evolving. This manuscript contributes by combining multiple drought indicators with an impact-based ML model, offering a novel approach.
- **Suggested Revision:** Provide a more explicit comparison with prior ML-based drought assessments to clarify how this approach improves upon past methods (e.g., better accuracy, broader applicability, or deeper insights into drought impacts).

Our study uses machine learning in two ways – first, we use machine learning to develop an impact-based drought metric that captures droughts using drought impact reports from local and federal government authorities. This approach builds upon previous studies that have used machine learning in combination with drought impact reports to assess the drivers and predictability of drought (Devanand et al., 2024; Hobeichi et al., 2022). Our approach goes

beyond previous research as we use the impact-based drought metric to quantify historical drought trends, complementing our drought trend analysis using more traditional drought metrics. In doing so, we show that traditional drought metrics are able to capture the historical trends in drought seen in the impact-based drought metric.

Secondly, we use machine learning to understand drivers of drought trends (i.e. which hydrometeorological variables contributed most to past changes in drought in each season and region). The application of machine learning in this context is novel and provides important insights into the most important drivers of drought trends.

We will edit the manuscript to detail the novelty and advancement over past approaches and studies, as suggested.

**3. Are substantial conclusions reached?**
- Yes, the manuscript draws key conclusions about the historical variability of droughts and suggests that recent trends are not unprecedented. However, the phrase "within historical variability" appears multiple times without clarifying whether it implies a lack of anthropogenic influence or if natural variability obscures a long-term trend.
- **Suggested Revision:** Add a clarifying statement in the discussion that "within variability" does not necessarily imply no human influence, as variability can mask emerging signals. Also, consider discussing whether these trends are expected to continue or if they are part of cyclical natural variability.

We agree with the reviewer that a drought trend being within the bounds of historical variability does not imply that there is no human influence. We were careful with the wording in our manuscript to ensure we stated "within observed variability" (as opposed to natural variability) as the historical changes can be due to both natural variability and anthropogenic influences.

We cannot quantify the contribution of the trends from human or natural influences separately with the data used in this study. An attribution study of this nature would typically rely on climate models, however these are not yet reliable for this purpose (Lane et al., 2023). We will revise the manuscript to explain that observed variability could include human and natural variability.

**Scientific Quality**
**4. Are the scientific methods and assumptions valid and clearly outlined?**
The methodology is well-described, but the manuscript could be improved with more explicit discussion of the following points:
- **Threshold Selection:** The authors justify the choice of the 15th percentile threshold, but should briefly discuss how this selection might impact results.

Past studies have found drought trends to be largely insensitive to the choice of threshold (e.g., Kirono et al., 2020; Ukkola et al., 2018). We expect that a lower percentile would likely lower the significance of trends due to a smaller sample size but would be unlikely to affect the sign of changes. We will revise the manuscript to clarify the choice of threshold and outline any potential impacts different threshold choices would have on the results.

- **Groundwater Use:** The limitations mention that AWRA-L does not account for groundwater use, but there is no discussion of how reliance on groundwater varies regionally and how this affects hydrological drought impacts.

We thank the reviewer for this suggestion and agree that this is an important point. Groundwater sources have large regional variation across Australia, and this could have influence over the impacts of drought. For example, areas with large groundwater reserves may have a delayed response to hydrological drought onset during extreme dry periods. On the other hand, groundwater can be linked to prolonged post-drought recovery, with catchments taking years to fully recover after long periods of drought (Fowler et al., 2020, 2022).

These responses to drought in regions with differing groundwater availability could have influence over the subsequent impacts of hydrological droughts. We will revise the manuscript to highlight the role of groundwater for hydrological drought impacts.

- **Drought Impact Data Bias:** The methodology explains that the RF model was trained on government drought reports, but it does not specify how those reports define or measure drought impacts. Furthermore, potential biases in training data due to regional population/economic factors should be acknowledged.

We agree it would be helpful to provide more details on how droughts were defined in the impact reports. We will include examples of how a drought impact was defined and acknowledge possible biases due to population and economic factors.

- **Uncertainty Consideration:** While variability in trends is discussed, the manuscript does not explicitly assess whether uncertainty bounds overlap with observed trends.

We are not sure we understand the reviewer's question, but we think it relates to whether observed trends are within the bounds of observed variability, i.e. if the trends are statistically significant in the context of observed variability.

Our study includes a comprehensive assessment of the emergence of drought trends from observed variability (see Section 2.4 and S1.2 for details of the methods). This includes a comparison of historical trends against uncertainty bounds of decadal means and the application of two trend emergence tests; the signal-to-noise (S/N) ratio and Kolmogorov-Smirnoff (KS) test (see the description of these results in Sections 3.1 and 3.2). We found that while some areas showed signs of trend emergence, the trends largely remain within the observed variability.

In summary, we believe that this comment is already addressed in the current version of the manuscript.

**5. Are the results sufficient to support the interpretations and conclusions?**
While the results are well presented, statistical transparency needs improvement. Specific concerns include:

- **Random Forest Model Overfitting:** The results discuss Mean Decrease in Impurity (MDI) for feature importance but do not report whether individual decision trees reached pure leaf outcomes (fully deterministic splits).
- **Regional Data Sparsity:** The manuscript does not explicitly address how the RF model handles regional data sparsity and whether feature importance rankings shift across different drought types and regions.
- **Suggested Revision**: Report maximum and mean tree depth and include a diagnostic plot of impurity reduction versus feature correlation. Additionally, summarize MDI variations between data-rich and datapoor regions and provide out-of-sample error distributions to assess model consistency.

We appreciate the reviewer's attention to the details of our Random Forest implementation. Please find our responses to each part of the suggested revisions below.

**Random Forest Model Overfitting**

We do not think that information on the purity of individual decision tree leaves contributes meaningfully to interpreting feature importance via Mean Decrease in Impurity (MDI). Due to the nature of bootstrap sampling and inherent randomness in feature selection in Random Forest models, it is likely that some trees may reach pure leaves simply by chance while others may not.

In our analysis, we employed the default settings of the `RandomForestClassifier` in the scikit-learn Python package, which allows trees to grow until no further impurity decrease is possible. This approach follows standard practice and is consistent with typical use of MDI.

**Regional Data Sparsity**

We believe there may have been a misunderstanding regarding the data used to train the Random Forest models for investigating drivers of drought trends. The training data are uniformly gridded datasets (gridded climate and hydrological data and gridded drought trends) and so have equal density across the regions. It is true that the underlying observations used to develop the gridded data have more density in some regions of Australia than in others. However, quantifying the effects of this on drought trends or feature importances is not within the scope of our study as it would require re-generating the climate data using a subset of stations, which is not a trivial undertaking. However, our manuscript already reports how the performance of the Random Forest and the feature importances vary between regions (see Table S1 for regionally and seasonally varying RF performance, and Figure 7 for regionally and seasonally varying feature importances).

**Out-of-sample error distributions**

All Random Forest performance metrics shown in our manuscript are based on out-of-sample predictions using a cross-validation approach (Section 2.2.2, lines 143-149, and Section 2.6, lines 234-240).

**6. Is the description of experiments and calculations sufficiently complete and precise to allow their reproduction by fellow scientists (traceability of results)?**
- The methodology is detailed, including data sources, statistical methods, and ML model parameters.

- **Suggested Revision:** However, some calculations (e.g., Mann-Kendall test and signal-to-noise ratio calculations) lack explicit mathematical definitions. Adding these formulae to the supplement would improve reproducibility.

We will modify the manuscript to include the signal-to-noise formula in the supplementary information. Python packages were used to calculate Mann-Kendall and Kolmogorov-Smirnoff tests. We will add citations of these libraries to the methods section.

**7. Do the authors give proper credit to related work and clearly indicate their own new/original contribution?**
- The manuscript references key studies on Australian drought trends and ML applications (e.g., at line 135, Devanand et al., 2024; Hobeichi et al., 2022).
- **Suggested Revision:** Strengthen the discussion on how this ML-based approach differs from previous studies. Explicitly highlight whether it provides higher accuracy, broader applicability, or novel insights into drought impacts.

As stated above, we will revise the manuscript to include this more explicit comparison as suggested (see our response to comment 2).

**Presentation Quality**
**8. Does the title clearly reflect the contents of the paper?**
- The title is relevant but could better emphasize the study's significance or approach.

**9. Does the abstract provide a concise and complete summary?**
- Yes, the abstract effectively outlines the research gap, methodology, key findings, and conclusions.

**10. Is the overall presentation well-structured and clear?**
Generally clear, but some linking between results and discussion needs improvement.
- Strengthen the explanation of seasonal drought trends by linking them explicitly to known climate drivers (ENSO, IOD, SAM).

We appreciate the reviewer's suggestion on strengthening the link between results and discussion and explaining seasonal drought trends by linking them to known climate drivers. Previous research has shown that seasonal climate drivers (such as ENSO, IOD, SAM) only explain a small proportion of Australian droughts (Hobeichi et al., 2024) and droughts are influenced by complex synoptic patterns and the frequency of heavy precipitation events (Holgate et al., 2025). As such we have decided not to explicitly link these climate drivers to seasonal drought trends in our analysis. Holgate et al. (2025) provide a review of meteorological droughts in Australia, and explain the connection with large-scale climate drivers in a more comprehensive manner than is possible within the scope of our study. In addition, Wasko et al. (2021) comprehensively discuss historical drivers of trends in hydrological variables and extremes.

- Clarify why evapotranspiration (ET) is a dominant drought driver in some regions but not others.

Differences in the most important drivers of drought trends reflect the differences between climate zones and hydrological regimes across Australia (e.g. the Monsoonal North with a distinct wet and dry season vs a temperate to dry climate in the South-East). We will expand

the discussion of these results to explain why the importance rankings of ET are as expected.

**11. Is the language fluent and precise?**
- Yes, the manuscript is well-written with clear and precise language.

**12. Are mathematical formulae, symbols, abbreviations, and units correctly defined and used?**
- Yes, but some statistical methods (e.g., Mann-Kendall test, signal-to-noise ratio calculations) lack explicit mathematical definitions.
- **Suggested Revision:** Include these formulae in supplementary materials for clarity, ensuring that readers unfamiliar with these methods can fully understand their application.

As stated above, we will include the necessary formulae and python package citations (see our response to comment 6).

**13. Should any parts of the paper (text, formulae, figures, tables) be clarified, reduced, combined, or eliminated?**
- Table S1 presents $R^2$ values for RF models, but it is not easy to compare performance across seasons/regions.
- **Suggested Revision:** Add a column with mean $R^2$ scores across all seasons/regions.

We agree that it would be useful to be able to compare the skill of the models across different regions and seasons. However, we believe taking a mean of the $R^2$ scores across the different random forest models would not be a meaningful statistic. Instead, we will add a column indicating the range of the $R^2$ scores for the different models.

**14. Are the number and quality of references appropriate?**
- Yes, but some references (e.g., Ukkola et al., 2024) appear incomplete.
- **Suggested Revision:** Ensure consistent formatting and completeness.

We will edit the references to ensure consistent formatting and completeness.

**15. Is the amount and quality of supplementary material appropriate?**
- The supplementary material is valuable, but some critical methodological details (e.g., preprocessing steps for government drought reports) should be included.

We will include these details as suggested by the reviewer in the above comments.

**References**

Devanand, A., Falster, G. M., Gillett, Z. E., Hobeichi, S., Holgate, C. M., Jin, C., Mu, M., Parker, T., Rifai, S. W., Rome, K. S., Stojanovic, M., Vogel, E., Abram, N. J., Abramowitz, G., Coats, S., Evans, J. P., Gallant, A. J. E., Pitman, A. J., Power, S. B., Rauniyar, S. P., Taschetto, A. S., and Ukkola, A. M.: Australia's Tinderbox Drought: An extreme natural event likely worsened by human-caused climate change, Sci. Adv., 10, eadj3460, https://doi.org/10.1126/sciadv.adj3460, 2024.

Fowler, K., Knoben, W., Peel, M., Peterson, T., Ryu, D., Saft, M., Seo, K.-W., and Western,

A.: Many Commonly Used Rainfall-Runoff Models Lack Long, Slow Dynamics: Implications for Runoff Projections, Water Resour. Res., 56, e2019WR025286, https://doi.org/10.1029/2019WR025286, 2020.

Fowler, K., Peel, M., Saft, M., Peterson, T. J., Western, A., Band, L., Petheram, C., Dharmadi, S., Tan, K. S., Zhang, L., Lane, P., Kiem, A., Marshall, L., Griebel, A., Medlyn, B. E., Ryu, D., Bonotto, G., Wasko, C., Ukkola, A., Stephens, C., Frost, A., Gardiya Weligamage, H., Saco, P., Zheng, H., Chiew, F., Daly, E., Walker, G., Vervoort, R. W., Hughes, J., Trotter, L., Neal, B., Cartwright, I., and Nathan, R.: Explaining changes in rainfall–runoff relationships during and after Australia's Millennium Drought: a community perspective, Hydrol. Earth Syst. Sci., 26, 6073–6120, https://doi.org/10.5194/hess-26-6073-2022, 2022.

Hobeichi, S., Abramowitz, G., Evans, J. P., and Ukkola, A.: Toward a Robust, Impact-Based, Predictive Drought Metric, Water Resour. Res., 58, e2021WR031829, https://doi.org/10.1029/2021WR031829, 2022.

Hobeichi, S., Abramowitz, G., Sen Gupta, A., Taschetto, A. S., Richardson, D., Rampal, N., Ayat, H., Alexander, L. V., and Pitman, A. J.: How well do climate modes explain precipitation variability?, Npj Clim. Atmospheric Sci., 7, 1–9, https://doi.org/10.1038/s41612-024-00853-5, 2024.

Holgate, C. M., Falster, G. M., Gillett, Z. E., Goswami, P., Grant, M. O., Hobeichi, S., Hoffmann, D., Jiang, X., Jin, C., Lu, X., Mu, M., Page, J. C., Parker, T. J., Vogel, E., Abram, N. J., Evans, J. P., Gallant, A. J. E., Henley, B. J., Kala, J., King, A. D., Maher, N., Nguyen, H., Pitman, A. J., Power, S. B., Rauniyar, S. P., Taschetto, A. S., and Ukkola, A. M.: Physical mechanisms of meteorological drought development, intensification and termination: an Australian review, Commun. Earth Environ., 6, 1–14, https://doi.org/10.1038/s43247-025-02179-3, 2025.

Kirono, D. G. C., Round, V., Heady, C., Chiew, F. H. S., and Osbrough, S.: Drought projections for Australia: Updated results and analysis of model simulations, Weather Clim. Extrem., 30, 100280, https://doi.org/10.1016/j.wace.2020.100280, 2020.

Lane, T. P., King, A. D., Perkins-Kirkpatrick, S. E., Pitman, A. J., Alexander, L. V., Arblaster, J. M., Bindoff, N. L., Bishop, C. H., Black, M. T., Bradstock, R. A., Clarke, H. G., Gallant, A. J. E., Grose, M. R., Holbrook, N. J., Holland, G. J., Hope, P. K., Karoly, D. J., Raupach, T. H., and Ukkola, A. M.: Attribution of extreme events to climate change in the Australian region – A review, Weather Clim. Extrem., 42, 100622, https://doi.org/10.1016/j.wace.2023.100622, 2023.

Ukkola, A. M., Pitman, A. J., Donat, M. G., De Kauwe, M. G., and Angélil, O.: Evaluating the Contribution of Land-Atmosphere Coupling to Heat Extremes in CMIP5 Models, Geophys. Res. Lett., 45, 9003–9012, https://doi.org/10.1029/2018GL079102, 2018.

Wasko, C., Shao, Y., Vogel, E., Wilson, L., Wang, Q. J., Frost, A., and Donnelly, C.: Understanding trends in hydrologic extremes across Australia, J. Hydrol., 593, 125877, https://doi.org/10.1016/j.jhydrol.2020.125877, 2021.

---

## Author Comment (AC2)

**Reviewer 2**

**Review of Grant et al., Historical trends of seasonal droughts in Australia**

The manuscript presents a statistical assessment of the historical trends of drought in Australia. The methodology and the analysis are well designed and contain elements of novelty, while the results are interesting, relevant and generally well discussed. There are, however, area where the manuscript could be improved, in particular with regards to the explanation of some of the methodologies and their contribution to the final results, as well as discussion of the implication of the statistical choices made.
Overall, the manuscript is of good quality and I recommend its publication in HESS with minor amendments.

We would like to sincerely thank the reviewer for taking the time to review the manuscript and for their positive assessment.

**Scientific significance:**
Does the manuscript represent a substantial contribution to scientific progress within the scope of Hydrology and Earth System Sciences (substantial new concepts, ideas, methods, or data)? **YES**

Despite presenting what is in practice a statistical analysis, the manuscript represents a significant contribution both methodologically and, to a lesser extent, because of the results presented. Methodologically, the manuscript presents a sound and thorough methodology for historical drought assessment, taking into consideration different aspects contributing to drought hazard, as well as potential impacts. The application of ML to model drought impact data is particularly interesting in this regard. Most of the results presented, additionally, highlight the complexity of drought as a natural disaster and present some interesting novel insight.

While the implications for agriculture and water supply of the results are outlined in the discussion section, the relevance of the manuscript could benefit from a deeper discussion of the relationship between the results obtained with the impact-based metric and the traditional ones and their implication. e.g. what meteorological variables seem to be the most relevant drivers of impact? What does this entail with regards to future climate scenarios?

We thank the reviewer for their suggestion. We agree that it would be helpful to more explicitly connect the trends in the impacts metric to the trends seen in the traditional metrics. We have shown that traditional drought metrics are able to capture historical trends in drought that we see in the impact-based drought metric (see Section 3.1), suggesting the traditional metrics are able to capture trends in drought impacts. We will expand further on the implications of the trends in the impacts-based metric on our understanding of the traditional metric trends.

**Scientific quality:**
Are the scientific approach and applied methods valid? Are the results discussed in an appropriate and balanced way (consideration of related work, including appropriate references)? **YES**

The methodology is well designed and generally well explained and referenced. Assumptions associated with the various methods are also generally well explained and reasoned. However, a few methodological choices could benefit for better explanation, reasoning and discussion of their implications:

- The choice of using the 15th percentile threshold as opposed to more traditional metrics, especially for the precipitation (SPI).

The 15th percentile is approximately equivalent to an SPI of -1 (Mckee et al., 1993), but using this method does not require any assumption of the data's distribution, and so can be applied consistently across the multiple drought types. We will revise the methods section to further clarify this.

- The trend emergence section (2,4) seems to have an implicit assumption of normality in the calculation of the Cis, is this correct? This approach feels like it needs better justification, the text just says that it is "used in previous studies".

We will revise the manuscript to include assumptions and limitations of the statistical methods used, where we have not already done so. Additionally, we will provide a more detailed explanation and justification of the decadal confidence intervals method used for assessing the stationarity of the area under drought trends.

- Also with regards to this section, I had to read through several times to understand how this was was differently for "time in drought" and "area in drought", could be more explicit.

The decadal confidence intervals were only applied to the area under drought trends, whereas the signal-to-noise and Kolmogorov-Smirnoff tests were applied to both the time and area under drought trends. We will review this section and, where necessary, make the text more explicit.

- Section 2.5 about the contributions from mean and variability seem quite simplistic and is not clear to be if this is a method that the authors came up with or is rooted in literature (there are no references). If the latter, a discussion of the limitations is warranted.

The method was developed by the authors and used here as there was no method in the literature which attributed drought trends to the mean and variability changes of the underlying hydrological variable. We will further clarify the method and its motivation and methodological justification in the revised manuscript.

- In general, in section 4.2 on the limitations, the focus is on the data, however a better discussion of the limitations and the assumptions of the statistical methods and tests performed should be included.

We have included limitations of the statistical methods within the results sections relevant to these methods. For example, we discuss the limitations and assumptions of the feature importance of the Random Forest models in Section 3.5 (L396-398). We will add information on assumptions and limitations of statistical tests where we have not already done so.

The results are also clear and well presented and their implications and limitations are well discussed, however the results from the RF model could benefit from additional synthesis and discussion.

As stated above, we will expand the discussion to include the implications of the (Random Forest) impact-based metric.

**Presentation quality:**
Are the scientific results and conclusions presented in a clear, concise, and well-structured way (number and quality of figures/tables, appropriate use of English language)? **YES**

Except for the few examples already mentioned above, the presentation is clear and easy to follow. Tables and Figures are clear and relevant.

**References**

Mckee, T. B., Doesken, N. J., and Kleist, J.: The Relationship of Drought Frequency and Duration to Time Scales, 8th Conf. Appl. Climatol., 179–184, 1993.

---

## Author Response (AR1)

**Reviewer 1**

**Review of Grant et al., Historical Trends of Seasonal Droughts in Australia**

This manuscript analyzes historical drought trends in Australia using multiple drought indicators, including meteorological, agricultural, hydrological, and impact-based metrics. It employs explainable machine learning to assess hydrometeorological drivers and the influence of variability versus mean changes. The study is well structured, the methodology is generally robust, and offers a valuable contribution to the understanding of historical drought trends in Australia. However, I have a several recommendations for improvement to enhance clarity, statistical transparency, and methodological rigor. This review evaluates the manuscript against HESS criteria, with recommendations to enhance its impact and alignment.

We would like to sincerely thank the reviewer for taking the time to review the manuscript and for their positive assessment. Please note, the line numbers stated in the responses are in reference to the line numbers in the version of the revised manuscript with track changes.

**Scientific Significance**

**1. Does the paper address relevant scientific questions within the scope of HESS?**

- Yes, the manuscript examines historical drought trends and their drivers, contributing to the understanding of water availability and hydrological variability. The study's integration of multiple drought indicators and machine learning techniques is relevant to advancing hydrological monitoring and data analysis. Additionally, its focus on drought impacts aligns with HESS's interest in hydrology's interaction with climate and society.
- Suggested Revision: The discussion could further highlight how the findings might support sustainable water resource management and decision-making in response to drought variability.

Thanks for this suggestion. We have expanded the discussion to highlight the need for sustainable water resource management in future to combat the impacts of drought (L484-488):

"Our study highlights the importance of considering both changes in the mean and variability of precipitation, soil moisture and runoff for drought trends. There is considerable uncertainty in future impacts of climate change on water availability (Fowler et al., 2022; Wasko et al., 2024), and adaptation to changes in both the mean and variability of hydrological variables is critical to ensure sustainable water resource management in the future."

**2. Does the paper present novel concepts, ideas, tools, or data?**

- Yes, ML integration with traditional drought assessment is emerging in Australia (line 135, citing Devanand et al., 2024; Hobeichi et al., 2022). While advancements exist, the field is evolving. This manuscript contributes by combining multiple drought indicators with an impact-based ML model, offering a novel approach.
- Suggested Revision: Provide a more explicit comparison with prior ML-based drought assessments to clarify how this approach improves upon past methods

(e.g., better accuracy, broader applicability, or deeper insights into drought impacts).

Our study uses machine learning in two ways – first, we use machine learning to develop an impact-based drought metric that captures droughts using drought impact reports from local and federal government authorities. This approach builds upon previous studies that have used machine learning in combination with drought impact reports to assess the drivers and predictability of drought (Devanand et al., 2024; Hobeichi et al., 2022). Our approach goes beyond previous research as we use the impact-based drought metric to quantify historical drought trends, complementing our drought trend analysis using more traditional drought metrics. In doing so, we show that traditional drought metrics are able to capture the historical trends in drought seen in the impact-based drought metric. We have revised the manuscript to explicitly show how our methodology builds on previous machine learning applications in drought research (L147-150):

"While past studies have combined impact reports and machine learning to assess the drivers and predictability of drought (Devanand et al., 2024; Hobeichi et al., 2022), our methodology goes beyond these by applying the impact-based metric to historical drought trends. This allows us to compare trends in the traditional and impact-based drought metrics."

Secondly, we use machine learning to understand drivers of drought trends (i.e. which hydrometeorological variables contributed most to past changes in drought in each season and region). We have revised the manuscript to highlight the novelty of applying machine learning in this context (L248-250):

"This approach provides a new application of machine learning to Australian droughts by using it to untangle the key drivers of drought trends, with similar methods previously used to understand the drivers of individual drought events (Devanand et al., 2024)."

**3. Are substantial conclusions reached?**

- Yes, the manuscript draws key conclusions about the historical variability of droughts and suggests that recent trends are not unprecedented. However, the phrase "within historical variability" appears multiple times without clarifying whether it implies a lack of anthropogenic influence or if natural variability obscures a long-term trend.
- Suggested Revision: Add a clarifying statement in the discussion that "within variability" does not necessarily imply no human influence, as variability can mask emerging signals. Also, consider discussing whether these trends are expected to continue or if they are part of cyclical natural variability.

We agree with the reviewer that a drought trend being within the bounds of historical variability does not imply that there is no human influence. We were careful with the wording in our manuscript to ensure we stated "within observed variability" (as opposed to natural variability) as the historical changes can be due to both natural variability and anthropogenic influences.

We cannot quantify the contribution of the trends from human and natural influences separately with the data used in this study. An attribution study of this nature would typically

rely on climate models; however these are not yet reliable for this purpose (Lane et al., 2023). We have revised the manuscript to clarify that our results do not necessarily imply there is no human influence in the observed drought trends (L304-308):

"However, this does not necessarily mean there is no human influence within these trends. The baseline period used in the tests includes both natural and anthropogenic influences. This, alongside the high natural variability of Australia's climate, can mask emerging anthropogenic signals. Our tests simply show that there is not yet evidence of a robust climate change signal based on the available data, but future monitoring of these trends is essential to determine if they are predominantly anthropogenically or naturally driven."

**Scientific Quality**

- **4.** Are the scientific methods and assumptions valid and clearly outlined? The methodology is well-described, but the manuscript could be improved with more explicit discussion of the following points:
  - Threshold Selection: The authors justify the choice of the 15th percentile threshold, but should briefly discuss how this selection might impact results.

Past studies have found drought trends to be largely insensitive to the choice of threshold (e.g., Kirono et al., 2020). We expect that a lower percentile would likely lower the significance of trends due to a smaller sample size but would be unlikely to affect the sign of changes. We have revised the manuscript to further justify the choice of threshold and outline any potential impacts different threshold choices would have on the results (L121-128):

"We chose to use the 15th percentile as the drought threshold as this is approximately equal to an SPI threshold of –1 (i.e. a "moderate" drought; Mckee et al., 1993). Other drought thresholds would be valid, but previous studies have shown that the drought trends are largely insensitive to the choice of threshold (Kirono et al., 2020). By using the15th percentile we ensure a large enough sample size of drought events to reliably calculate trends; lower thresholds would likely give fewer significant trends but be unlikely to affect the sign of the change. By using an empirical percentile method, no assumption of a specific statistical distribution of the hydrological variable is required, which allows for a consistent methodology across the three traditional drought metrics."

• **Groundwater Use:** The limitations mention that AWRA-L does not account for groundwater use, but there is no discussion of how reliance on groundwater varies regionally and how this affects hydrological drought impacts.

We thank the reviewer for this suggestion and agree that this is an important point. Groundwater sources have large regional variation across Australia, and this could have influence over the impacts of drought. For example, areas with large groundwater reserves may have a delayed response to hydrological drought onset during extreme dry periods. On the other hand, groundwater can be linked to prolonged post-drought recovery, with catchments taking years to fully recover after long periods of drought (Fowler et al., 2020, 2022).

These responses to drought in regions with differing groundwater availability could have influence over the subsequent impacts of hydrological droughts. We have revised the manuscript to highlight the role of groundwater for hydrological drought impacts (L503-508):

"Additionally, AWRA-L does not account for human use of reservoir or aquifer water and is generally not suited to model complex groundwater to surface water interactions. However, regional variation in groundwater availability across Australia could influence drought onset and post-drought recovery which may not be captured in our study. For example, areas with large groundwater reserves would have a delayed onset of hydrological drought (Mu et al., 2022). At the same time, the influence of groundwater also modulates post-drought recovery, with some regions taking years to fully recover from drought due to delayed groundwater recharge (Fowler et al., 2020, 2022)."

• **Drought Impact Data Bias:** The methodology explains that the RF model was trained on government drought reports, but it does not specify how those reports define or measure drought impacts. Furthermore, potential biases in training data due to regional population/economic factors should be acknowledged.

We agree it would be helpful to provide more details on how droughts were defined in the impact reports. We have included examples of how a drought impact was defined and acknowledge possible biases due to population and economic factors (L139-142):

"Examples of the observed drought impacts include crops being grazed or cut for hay or silage, reported effects to water supply in major towns or cities, or inadequate water availability in the main storage dam. Given the nature of these reports, it should be noted that they may be biased towards large population or agricultural regions."

Uncertainty Consideration: While variability in trends is discussed, the
manuscript does not explicitly assess whether uncertainty bounds overlap with
observed trends.

We are not sure we understand the reviewer's question, but we think it relates to whether observed trends are within the bounds of observed variability, i.e. if the trends are statistically significant in the context of observed variability.

Our study includes a comprehensive assessment of the emergence of drought trends from observed variability (see Section 2.4 and S1.2 for details of the methods). This includes a comparison of historical trends against uncertainty bounds of decadal means and the application of two trend emergence tests; the signal-to-noise (S/N) ratio and Kolmogorov-Smirnoff (KS) test (see the description of these results in Sections 3.1 and 3.2). We found that while some areas showed signs of trend emergence, the trends largely remain within the observed variability.

In summary, we believe that this comment is already addressed in the current version of the manuscript.

**5.** Are the results sufficient to support the interpretations and conclusions? While the results are well presented, statistical transparency needs improvement. Specific concerns include:

- Random Forest Model Overfitting: The results discuss Mean Decrease in Impurity (MDI) for feature importance but do not report whether individual decision trees reached pure leaf outcomes (fully deterministic splits).
- Regional Data Sparsity: The manuscript does not explicitly address how the RF model handles regional data sparsity and whether feature importance rankings shift across different drought types and regions.
- Suggested Revision: Report maximum and mean tree depth and include a
  diagnostic plot of impurity reduction versus feature correlation. Additionally,
  summarize MDI variations between data-rich and datapoor regions and provide
  out-of-sample error distributions to assess model consistency.

We appreciate the reviewer's attention to the details of our Random Forest implementation. Please find our responses to each part of the suggested revisions below.

**Random Forest Model Overfitting**

We do not think that information on the purity of individual decision tree leaves contributes meaningfully to interpreting feature importance via Mean Decrease in Impurity (MDI). Due to the nature of bootstrap sampling and inherent randomness in feature selection in Random Forest models, it is likely that some trees may reach pure leaves simply by chance while others may not.

In our analysis, we employed the default settings of the RandomForestClassifier in the scikit-learn Python package, which allows trees to grow until no further impurity decrease is possible. This approach follows standard practice and is consistent with typical use of MDI.

**Regional Data Sparsity**

We believe there may have been a misunderstanding regarding the data used to train the Random Forest models for investigating drivers of drought trends. The training data are uniformly gridded datasets (gridded climate and hydrological data and gridded drought trends) and so have equal density across the regions. It is true that the underlying observations used to develop the gridded data have more density in some regions of Australia than in others. However, quantifying the effects of this on drought trends or feature importances is not within the scope of our study as it would require re-generating the climate data using a subset of stations, which is not a trivial undertaking. However, our manuscript already reports how the performance of the Random Forest and the feature importances vary between regions (see Table S1 for regionally and seasonally varying RF performance, and Figure 7 for regionally and seasonally varying feature importances).

**Out-of-sample error distributions**

All Random Forest performance metrics shown in our manuscript are based on out-of-sample predictions using a cross-validation approach (Section 2.2.2, L152-158, and Section 2.6, L 252-258).

6. Is the description of experiments and calculations sufficiently complete and precise to allow their reproduction by fellow scientists (traceability of results)?

 The methodology is detailed, including data sources, statistical methods, and ML model parameters. • Suggested Revision: However, some calculations (e.g., Mann-Kendall test and signal-to-noise ratio calculations) lack explicit mathematical definitions. Adding these formulae to the supplement would improve reproducibility.

We have now included the formula for the signal-to-noise ratio and information on the python packages used in its calculation in the supplementary information (L766-771). Additionally, we have added details of the python packages used for calculating the Mann-Kendall (L195-196) and Kolmogorov-Smirnoff tests (L775-776).

**7. Do the authors give proper credit to related work and clearly indicate their own new/original contribution?**

- The manuscript references key studies on Australian drought trends and ML applications (e.g., at line 135, Devanand et al., 2024; Hobeichi et al., 2022).
- **Suggested Revision:** Strengthen the discussion on how this ML-based approach differs from previous studies. Explicitly highlight whether it provides higher accuracy, broader applicability, or novel insights into drought impacts.

As stated above, we have revised the manuscript to include this more explicit comparison as suggested (see our response to comment 2).

**Presentation Quality**

- 8. Does the title clearly reflect the contents of the paper?
  - The title is relevant but could better emphasize the study's significance or approach.

Thank you for the suggestion. After further consideration, we have chosen to keep the current title of the paper as it concisely captures the focus of our study.

**9. Does the abstract provide a concise and complete summary?**

 Yes, the abstract effectively outlines the research gap, methodology, key findings, and conclusions.

**10. Is the overall presentation well-structured and clear?**

Generally clear, but some linking between results and discussion needs improvement.

We have made slight changes to the format of the Sections to make a clearer link between discussions and results. We have combined Results and Discussion into a single section (Section 4) to discuss our results directly after they are presented and include some overarching discussion points at the end of Section 4. Additionally, we have moved Key Messages to its own section (Section 5).

• Strengthen the explanation of seasonal drought trends by linking them explicitly to known climate drivers (ENSO, IOD, SAM).

We appreciate the reviewer's suggestion on strengthening the link between results and discussion and explaining seasonal drought trends by linking them to known climate drivers. Previous research has shown that seasonal climate drivers (such as ENSO, IOD, SAM) only explain a small proportion of Australian droughts (Hobeichi et al., 2024) and droughts are influenced by complex synoptic patterns and the frequency of heavy precipitation events

(Holgate et al., 2025). As such we have decided not to explicitly link these climate drivers to seasonal drought trends in our analysis. Holgate et al. (2025) provide a review of meteorological droughts in Australia and explain the connection with large-scale climate drivers in a more comprehensive manner than is possible within the scope of our study.

• Clarify why evapotranspiration (ET) is a dominant drought driver in some regions but not others.

Differences in the most important drivers of drought trends reflect the differences between climate zones and hydrological regimes across Australia (e.g. the Monsoonal North with a distinct wet and dry season vs a temperate to dry climate in the South-East). We have revised the manuscript to explain why ET ranks as the most important variable in the Rangelands, least important in the Wet Tropics and varies seasonally in regions with distinct seasons (L436-442):

"However, this varies greatly between the different regions; for example, the mean trend in ET consistently ranks as the least important hydrometeorological variable in the Wet Tropics across all the seasons. This is likely due to the year-round wet conditions in this region, leading to smaller variations in ET and thus a limited influence on droughts. Conversely, ET has a larger influence in regions and seasons where it has a high contribution on the water cycle. For example, ET consistently ranks as the most important variable in the precipitation-limited Rangelands region. Regions with more distinct seasons such as southeastern regions or the Monsoonal North have greater variations in the importance of ET across the seasons. For example, in the Southern Slopes the most important predictor is the runoff mean trend in DJF and JJA, ET mean trend in MAM, and precipitation standard deviation trend in SON."

**11. Is the language fluent and precise?**

• Yes, the manuscript is well-written with clear and precise language.

**12. Are mathematical formulae, symbols, abbreviations, and units correctly defined and used?**

- Yes, but some statistical methods (e.g., Mann-Kendall test, signal-to-noise ratio calculations) lack explicit mathematical definitions.
- Suggested Revision: Include these formulae in supplementary materials for clarity, ensuring that readers unfamiliar with these methods can fully understand their application.

Thank you for this comment. We have included the necessary formulae and details of python packages used (see our response to comment 6).

**13. Should any parts of the paper (text, formulae, figures, tables) be clarified, reduced, combined, or eliminated?**

- Table S1 presents R² values for RF models, but it is not easy to compare performance across seasons/regions.
- Suggested Revision: Add a column with mean R2 scores across all seasons/regions.

We agree that it would be useful to be able to compare the skill of the models across different regions and seasons. However, we believe taking a mean of the R2 scores across

the different random forest models would not be a meaningful statistic. Instead, we have added columns to Table S1 to indicate the range of R2 scores for each region, and a row to indicate the range for each season.

**14. Are the number and quality of references appropriate?**

- Yes, but some references (e.g., Ukkola et al., 2024) appear incomplete.
- Suggested Revision: Ensure consistent formatting and completeness.

We have edited the references to ensure consistent formatting and completeness.

**15. Is the amount and quality of supplementary material appropriate?**

• The supplementary material is valuable, but some critical methodological details (e.g., preprocessing steps for government drought reports) should be included.

We have included these details as suggested by the reviewer in the above comments.

**Reviewer 2**

**Review of Grant et al., Historical trends of seasonal droughts in Australia**

The manuscript presents a statistical assessment of the historical trends of drought in Australia. The methodology and the analysis are well designed and contain elements of novelty, while the results are interesting, relevant and generally well discussed. There are, however, area where the manuscript could be improved, in particular with regards to the explanation of some of the methodologies and their contribution to the final results, as well as discussion of the implication of the statistical choices made.

Overall, the manuscript is of good quality and I recommend its publication in HESS with minor amendments.

We would like to sincerely thank the reviewer for taking the time to review the manuscript and for their positive assessment. Please note, the line numbers stated in the responses are in reference to the line numbers in the version of the revised manuscript with track changes.

**Scientific significance:**

Does the manuscript represent a substantial contribution to scientific progress within the scope of Hydrology and Earth System Sciences (substantial new concepts, ideas, methods, or data)? **YES**

Despite presenting what is in practice a statistical analysis, the manuscript represents a significant contribution both methodologically and, to a lesser extent, because of the results presented. Methodologically, the manuscript presents a sound and thorough methodology for historical drought assessment, taking into consideration different aspects contributing to drought hazard, as well as potential impacts. The application of ML to model drought impact data is particularly interesting in this regard. Most of the results presented, additionally, highlight the complexity of drought as a natural disaster and present some interesting novel insight.

While the implications for agriculture and water supply of the results are outlined in the discussion section, the relevance of the manuscript could benefit from a deeper discussion of the relationship between the results obtained with the impact-based metric and the traditional ones and their implication. e.g. what meteorological variables seem to be the most relevant drivers of impact? What does this entail with regards to future climate scenarios?

We thank the reviewer for their suggestion. We agree that it would be helpful to more explicitly connect the trends in the impacts metric to the trends seen in the traditional metrics. We have shown that traditional drought metrics are able to capture historical trends in drought that we see in the impact-based drought metric (see Section 3.1), suggesting the traditional metrics are able to capture trends in drought impacts. We have expanded further on the implications of the trends in the impacts-based metric on our understanding of the traditional metric trends in L147-150:

"While past studies have combined impact reports and machine learning to assess the drivers and predictability of drought (Devanand et al., 2024; Hobeichi et al., 2022), our methodology goes beyond these by applying the impact-based metric to historical drought

trends. This allows us to compare trends in the traditional and impact-based drought metrics."

And, additionally in L316-317:

"This implies that the physical changes seen in the traditional metrics are likely mirrored in changes to drought impacts."

**Scientific quality:**

Are the scientific approach and applied methods valid? Are the results discussed in an appropriate and balanced way (consideration of related work, including appropriate references)? **YES**

The methodology is well designed and generally well explained and referenced. Assumptions associated with the various methods are also generally well explained and reasoned. However, a few methodological choices could benefit for better explanation, reasoning and discussion of their implications:

• The choice of using the 15th percentile threshold as opposed to more traditional metrics, especially for the precipitation (SPI).

The 15th percentile is approximately equivalent to an SPI of -1 (Mckee et al., 1993), but unlike SPI, the percentile method does not require any assumption of the data's distribution and so can be applied consistently across the multiple drought types. We have revised the methods section to clarify this point (L121-128):

"We chose to use the 15th percentile as the drought threshold as this is approximately equal to an SPI threshold of –1 (i.e. a "moderate" drought; Mckee et al., 1993). Other choices of thresholds would be valid, but previous studies have shown that the drought trends are largely insensitive to the choice of threshold (Kirono et al., 2020). By using the15th percentile we ensure a large enough sample size of drought events to reliably calculate trends; lower thresholds would likely give fewer significant trends but be unlikely to affect the sign of the change. By using an empirical percentile method, no assumption of a specific statistical distribution of the hydrological variable is required, which allows for a consistent methodology across the three traditional drought metrics."

• The trend emergence section (2,4) seems to have an implicit assumption of normality in the calculation of the Cis, is this correct? This approach feels like it needs better justification, the text just says that it is "used in previous studies".

Thank you for this reply. We would like to note that we do not calculate the 95% interval of area under drought values, but the 95% confidence interval of the decadal mean of area under drought values. In other words, we are calculating the 95% interval of the estimates of the mean. According to the central limit theorem, estimates of a mean of a population are normally distributed even if the underlying population is not normally distributed. So, the implicit assumption in Eq. 3 is that the estimates of the mean of area under drought are normally distributed, which holds. We have added this clarification in L214:

"And finally, from this we could find the confidence intervals of the decadal means of area under drought with:"

Additionally, we have provided a more detailed explanation and justification of this method (L205-209):

"Firstly, for area under drought trends, we analysed the changes in decadal means to identify long-term and lasting shifts in area under drought for each NRM region. If shifts have occurred, this method allows for the identification of the timing of the shift. Adopting the methods used in previous studies (Sun et al., 2018; Ukkola et al., 2019), 95% confidence intervals were calculated for the decadal means"

 Also with regards to this section, I had to read through several times to understand how this was was differently for "time in drought" and "area in drought", could be more explicit.

The decadal confidence intervals were only applied to the area under drought trends, whereas the signal-to-noise and Kolmogorov-Smirnoff tests were applied to both the time and area under drought trends. We have revised the manuscript to make this clearer when it is described in Section 2.4 (L221) and Section 3.2 (L351-352):

"The method for detecting changes in decadal means was only applied to the area under drought trends."

"For the area under drought trends, we assessed the trend emergence using three methods: changes in decadal means, KS test, and the S/N ratio."

 Section 2.5 about the contributions from mean and variability seem quite simplistic and is not clear to be if this is a method that the authors came up with or is rooted in literature (there are no references). If the latter, a discussion of the limitations is warranted.

The method was developed by the authors and used here as there was no method in the literature which attributed drought trends to the mean and variability changes of the underlying hydrological variable. We have clarified this point in the methods and detailed the motivation and methodological justification in the revised manuscript (L225-226):

"We developed and applied a new methodology to isolate the contributions of mean and variability changes. Despite its simplicity, it offers further insight into the underlying causes of the drought trends."

• In general, in section 4.2 on the limitations, the focus is on the data, however a better discussion of the limitations and the assumptions of the statistical methods and tests performed should be included.

In the initial submission of the manuscript, we had included limitations of some of the statistical methods within the results sections relevant to these methods. For example, we discuss the limitations and assumptions of the feature importance of the Random Forest models in Section 3.5 (L421-423). Many of the other statistical tests used in our analysis (Mann-Kendall, Kolmogorov-Smirnoff, and signal to noise ratio) are non-parametric tests and so do not require assumptions about the underlying data.

We have revised the manuscript to include possible limitations of the Mann-Kendall trend test (L190-194):

"This method uses a pre-whitening process to deal with autocorrelation. In doing so, there is a chance it reduces the power of the trend test due to the pre-whitening process potentially removing trend information in cases where the autocorrelation and trend are intrinsically linked (Yue and Wang, 2004). This means that the method is conservative with borderline significant trends and reduces the likelihood of over-estimating the trend significance."

As well as possible limitations that arise from our implementation of the signal-to-noise ratio and Kolmogorov-Smirnoff test (L758-762):

"For both methods the first 50 years (1911-1961) was used as a baseline to compare emergence to. Due to data availability and the importance of using a long enough baseline period to capture the variability, this is the earliest baseline period we can use. However, it should be noted that this baseline already likely includes anthropogenic forcings within it, therefore it is possible that the results from these tests underestimate or misrepresent the trend emergence from the natural variability. Due to this, we refer to the trend emergence as emergence from the observed or historic variability so that it is clear we are not implying emergence from natural variability."

The results are also clear and well presented and their implications and limitations are well discussed, however the results from the RF model could benefit from additional synthesis and discussion.

Thank you for this comment. As stated above, we have expanded the discussion to include the implications of the (Random Forest) impact-based metric.

**Presentation quality:**

Are the scientific results and conclusions presented in a clear, concise, and well-structured way (number and quality of figures/tables, appropriate use of English language)? **YES**

Except for the few examples already mentioned above, the presentation is clear and easy to follow. Tables and Figures are clear and relevant.

**References**

Devanand, A., Falster, G. M., Gillett, Z. E., Hobeichi, S., Holgate, C. M., Jin, C., Mu, M., Parker, T., Rifai, S. W., Rome, K. S., Stojanovic, M., Vogel, E., Abram, N. J., Abramowitz, G., Coats, S., Evans, J. P., Gallant, A. J. E., Pitman, A. J., Power, S. B., Rauniyar, S. P., Taschetto, A. S., and Ukkola, A. M.: Australia's Tinderbox Drought: An extreme natural event likely worsened by human-caused climate change, Sci. Adv., 10, eadj3460, https://doi.org/10.1126/sciadv.adj3460, 2024.

Fowler, K., Knoben, W., Peel, M., Peterson, T., Ryu, D., Saft, M., Seo, K.-W., and Western, A.: Many Commonly Used Rainfall-Runoff Models Lack Long, Slow Dynamics: Implications for Runoff Projections, Water Resour. Res., 56, e2019WR025286, https://doi.org/10.1029/2019WR025286, 2020.

- Fowler, K., Peel, M., Saft, M., Nathan, R., Horne, A., Wilby, R., McCutcheon, C., and Peterson, T.: Hydrological Shifts Threaten Water Resources, Water Resour. Res., 58, e2021WR031210, https://doi.org/10.1029/2021WR031210, 2022.
- Hobeichi, S., Abramowitz, G., Evans, J. P., and Ukkola, A.: Toward a Robust, Impact-Based, Predictive Drought Metric, Water Resour. Res., 58, e2021WR031829, https://doi.org/10.1029/2021WR031829, 2022.
- Kirono, D. G. C., Round, V., Heady, C., Chiew, F. H. S., and Osbrough, S.: Drought projections for Australia: Updated results and analysis of model simulations, Weather Clim. Extrem., 30, 100280, https://doi.org/10.1016/j.wace.2020.100280, 2020.
- Mckee, T. B., Doesken, N. J., and Kleist, J.: The Relationship of Drought Frequency and Duration to Time Scales, 8th Conf. Appl. Climatol., 179–184, 1993.
- Mu, M., Pitman, A. J., De Kauwe, M. G., Ukkola, A. M., and Ge, J.: How do groundwater dynamics influence heatwaves in southeast Australia?, Weather Clim. Extrem., 37, 100479, https://doi.org/10.1016/j.wace.2022.100479, 2022.
- Sun, F., Roderick, M. L., and Farquhar, G. D.: Rainfall statistics, stationarity, and climate change, Proc. Natl. Acad. Sci., 115, 2305–2310, https://doi.org/10.1073/pnas.1705349115, 2018.
- Ukkola, A. M., Roderick, M. L., Barker, A., and Pitman, A. J.: Exploring the stationarity of Australian temperature, precipitation and pan evaporation records over the last century, Environ. Res. Lett., 14, 124035, https://doi.org/10.1088/1748-9326/ab545c, 2019.
- Wasko, C., Stephens, C., Peterson, T. J., Nathan, R., Pepler, A., Hettiarachchi, S., Vogel, E., Johnson, F., and Westra, S.: Understanding the implications of climate change for Australia's surface water resources: Challenges and future directions, J. Hydrol., 645, 132221, https://doi.org/10.1016/j.jhydrol.2024.132221, 2024.
- Yue, S. and Wang, C.: The Mann-Kendall Test Modified by Effective Sample Size to Detect Trend in Serially Correlated Hydrological Series, Water Resour. Manag., 18, 201–218, https://doi.org/10.1023/B:WARM.0000043140.61082.60, 2004.